# Stability of Human Telomeric G-Quadruplexes Complexed with Photosensitive Ligands and Irradiated with Visible Light

**DOI:** 10.3390/ijms24109090

**Published:** 2023-05-22

**Authors:** Valeria Libera, Francesca Ripanti, Caterina Petrillo, Francesco Sacchetti, Javier Ramos-Soriano, Maria Carmen Galan, Giorgio Schirò, Alessandro Paciaroni, Lucia Comez

**Affiliations:** 1Department of Physics and Geology, University of Perugia, Via Alessandro Pascoli, 06123 Perugia, Italy; valeria.libera@studenti.unipg.it (V.L.);; 2Istituto Officina dei Materiali-IOM, National Research Council-CNR, Via Alessandro Pascoli, 06123 Perugia, Italy; 3School of Chemistry, University of Bristol, Cantock’s Close, Bristol BS8 1TS, UK; 4CNRS, CEA, IBS, c/o University Grenoble Alpes, 38400 Grenoble, France

**Keywords:** G-quadruplex, singular value decomposition, photosensitive ligands, circular dichroism

## Abstract

Guanine-rich DNA sequences can fold into non-canonical nucleic acid structures called G-quadruplexes (G4s). These nanostructures have strong implications in many fields, from medical science to bottom-up nanotechnologies. As a result, ligands interacting with G4s have attracted great attention as candidates in medical therapies, molecular probe applications, and biosensing. In recent years, the use of G4-ligand complexes as photopharmacological targets has shown significant promise for developing novel therapeutic strategies and nanodevices. Here, we studied the possibility of manipulating the secondary structure of a human telomeric G4 sequence through the interaction with two photosensitive ligands, DTE and TMPyP4, whose response to visible light is different. The effect of these two ligands on G4 thermal unfolding was also considered, revealing the occurrence of peculiar multi-step melting pathways and the different attitudes of the two molecules on the quadruplex stabilization.

## 1. Introduction

The design and development of customizable functional materials capable of carrying out controlled structural modifications upon activation with a photonic stimulus are essential for photon-fueled therapeutic and imaging applications. As a spatiotemporally controllable, noninvasive tool for biological applications, light presents unmatched perspectives. To date, the requirement to induce switching in at least one direction by UV-Visible (UV-Vis) light, which is frequently harmful and penetrates most media only partially, restrains the development of effective photodynamic systems in significant ways. For this reason, work is being performed in a variety of fields to individuate natural compounds or to create artificial photosensitive systems that react predominantly to visible and near-infrared light (400–1000 nm) [1].

A lot of naturally existing systems, including nucleic acids and proteins complexed with photosensitive ligands, have been used as structural platforms to design stimuli-responsive nanostructured hybrid materials with adaptable and reversible capabilities [2,3]. Among them, guanine (G)-rich nucleic acid sequences, able to form G-quadruplex (G4) motifs, are promising tunable metal-mediated assemblies that can be used in a wide range of applications, including molecular computing and biosensing [4,5]. G4s also attracted significant attention as therapeutic targets owing to their occurrence in human oncogene promoter sequences and the genomes of pathogenic organisms [6,7,8,9,10]. Despite the properties related to the medical field, the possibility of using G4-ligand complexes as photopharmacological targets is still challenging [2,11]. Particular interest in perspective could be found in cancer treatment, where G4s have been identified as possible targets for photodynamic therapy [12,13], which operates thanks to the combined use of a photosensitizing agent, a light source, and molecular oxygen to specifically destroy malignant cells. G4s may represent an ideal subject for this purpose, as they are very versatile structures whose topology can be triggered by various factors (e.g., metal cations, pH, crowding, and ligand-binding) [14,15]. G4s are scaffolds formed by stacking two or more planar G-quartets, where each G-quartet is made up of four G-bases that arrange cyclically via Hoogsteen hydrogen bonds. The glycosidic bond angle of the nucleobases can assume *anti* or *syn* dispositions [16], and different combinations of these units, i.e., *anti-anti*, *syn-anti*, *anti-syn*, or *syn-syn*, give rise to different folding topologies resulting in parallel and antiparallel strand orientations [17,18,19]. Hybrid conformations consisting of both parallel/antiparallel strands may also coexist [20,21], and several classifications have been reported in the literature, highlighting the polymorphism of G4s [21,22]. A cartoon of different G4 topologies is shown in Figure 1. By exploiting this polymorphic nature, G4-interacting photosensitive ligands can trigger topological variations on demand using different wavelengths and/or exposure times of the light source [11].

The present work is focused on human telomeric G4s, formed by tandem repeats of the sequence d(TTAGGG), which have been revealed in vivo in the terminal part of chromosomes [8]. Among them, we selected the AG3(TTAG3)3 (Tel22) sequence, which can exist in vitro as parallel, hybrid-3+1, antiparallel, and two-tetrad antiparallel (hybrid-3) structures [23,24,25,26,27,28,29], depending on various ionic and crowding conditions. Based on past literature, it appears that hybrid-type telomere conformations are the most physiologically and thermodynamically relevant [30]. Recent experiments confirm that Tel22 spontaneously forms hybrid-1, -2, and -3, but not the parallel or antiparallel basket topologies when injected into live HeLa cells [30]. Hence, to better mimic cellular conditions, Tel22 is being studied in a KCl solution, which promotes folding towards a mixture of topologies, including hybrid conformations [31,32]. Despite making Tel22 a more complex system, this choice represents a significant stride towards exploring physiological systems. Nevertheless, resolving the Tel22 structure in these K^+^ environmental conditions via high-resolution techniques is particularly challenging. For instance, while Nuclear Magnetic Resonance (NMR) has proven to be a valuable tool for elucidating the G4 structure alone and in complex with ligands in solution [33,34,35], this approach faces limitations when studying Tel22 in a K^+^ environment due to the spectral overlap of different conformations. In this scenario, Circular Dichroism (CD) data and their refined analysis offer valuable insights into the secondary structures of complex systems with non-unique topologies.

In particular, we consider two photosensitizers targeting telomeric sequences and show the effects of their interaction with Tel22 on its secondary structure topology upon irradiation with visible light. The first ligand is a recently synthesized pyridinium-decorated dithienylethene (DTE) ligand, which has been shown to selectively target G4 with discrimination against duplex DNA and to produce reversible photoswitching between the open (1o) and closed (1c) isomers by alternating blue/red visible light exposure [2,11]. The second is TMPyP4 porphyrin, which is known for its ability to downregulate gene expression through quadruplex formation or induction in the promoter region [36]. Porphyrins produce singlet oxygen (^1^O_2_) when exposed to radiation [37], a feature often employed in photodynamic therapy [38]. The interaction between TMPyP4 and human telomeric G4 is widely studied, but it continues to be controversial [39]. DTE and TMPyP4 represent a model strategy for two future perspectives: (i) the design of photosensitive ligands to bind selectively to G4 telomeres in their light-activated state and dissociate in their dark state; (ii) the possibility to exploit photodynamic therapy, where photosensitive ligands can be targeted to cancer cells that have elevated telomerase activity, leading to specific damage or cell death upon light activation.

In this context, we used UV-Vis absorption and CD techniques to investigate the optical performance and chiral properties of Tel22–DTE and Tel22–TMPyP4 systems once irradiated. The secondary structure of native and photo-treated complexes is also analyzed as a function of temperature by applying the singular value decomposition (SVD) method, interestingly revealing the presence of intermediate conformational rearrangements before thermal unfolding. Besides assessing the stability of the complexes, using temperature as a controller enables exploration of the conformational landscape of Tel22-based systems, even in the presence of significant structural fluctuations, such as those found in crowded and cellular environments.

Overall, our findings corroborate the evidence that Tel22–photoligand complexes are well suited to photoregulation of G4 secondary structure, opening up a variety of possible uses in nanodevices and the chance to regulate G4 structure in biological contexts.

## 2. Results

### 2.1. Light as a Driving Factor for Conformational Changes in G4

Light irradiation was used to control the photochromic properties of the DTE and TMPyP4 ligands. When illuminating, the irradiation time is strictly correlated with the power density (power/area) of the light spot on the sample. To maintain mechanical stability while irradiating and ensure uniform illumination of the samples during the experiments, we utilized a custom-made irradiation setup (see Section 4 for details and Appendix A).

#### 2.1.1. Tel22–DTE Complex

DTE was selected because it has been shown to undergo reversible photoswitching between the open (1o) and closed (1c) isomers through alternate irradiation with blue and red visible light [2]. The samples were prepared based on the optimal stoichiometric ratio of ligands to DNA (2:1), as determined by titration experiments. The binding constant for both Tel22–DTE (1c) and Tel22–DTE (1o) samples was determined through the procedure described in Ref. [2] and found to be in the micromolar range. Using a system equipped with two LED lamps emitting at 430 nm and 660 nm, respectively, we first optimized the experimental conditions for photo-switching (irradiation time and power density), as shown in Figure 2.

The required exposure time for (1c) → (1o) switching is 1.5 h (660 nm), while for (1o) → (1c), the transition is 10 min (430 nm). Both forms (i.e., 1c and 1o) are stable after irradiation. Then, we checked the (1o) ⇄ (1c) reversibility over several (~10) illumination cycles, as reported in Appendix A. The overall absorption spectral signature of DTE (1o) presents two main peaks at about 329 nm and 380 nm, while the (1c) configuration has characteristic peaks at about 436 nm and 670 nm. The differences in UV-Vis absorption spectra provide a clear indication of unique electronic configurations, which may translate into various interactions with Tel22.

To further explore the different isomer properties, Tel22 was complexed with both DTE (1o) and DTE (1c). The CD spectra reported in Figure 3 provide spectral signatures of the Tel22-DTE interaction.

In particular, the peak intensity of the CD signal of the Tel22–DTE (1o) complex increases at 290 nm, and the overall spectrum between 230 and 260 nm significantly decreases, indicating a rearrangement of the Tel22 secondary structure. The G4-ligand interaction is also confirmed by the presence of the induced CD signal of the ligand in the 310–350 nm region. Indeed, interactions between chiral (Tel22) and achiral molecules (DTE) can give rise to induced CD of the achiral compound, related to its UV-Vis absorption. To better highlight the presence of this additional signal, we performed CD titration measurements by gradually increasing the amount of the DTE (1o) ligand and keeping the amount of Tel22 fixed. The resulting spectra are shown in Appendix A. The Tel22–DTE (1c) complex likewise undergoes similar structural changes, with slight differences in intensity. Interestingly, the effect induced by the DTE (1o) is quite different from what was previously observed in another telomeric sequence (Tel23) [2], where an overall decrease in CD spectra was attributed to a disruption of the native G-tetrad network based on NMR measurements. Unlike Tel23, the structure of Tel22 remains folded even after complexation with both isomers and subsequent illumination. This is supported by Small Angle X-rays Scattering (SAXS) experiments, which reveal the presence of a compact structure (see Appendix A).

To give an estimate of the irradiation effect on the complex and to check its reversibility, the Tel22–DTE (1o) sample was then exposed to blue light to allow ligand photo-isomerization, namely Tel22–DTE (1c’), which was monitored and confirmed through UV-Vis absorption measurements. A schematic representation of the process is shown in Figure 4a. Upon irradiation, the complex displayed a general increase in the CD intensity in the 230–280 nm range and the disappearance of the induced CD signal, resulting in a spectrum very similar, but not fully superimposable, to that of Tel22 directly complexed with DTE (1c). The Tel22–DTE (1c’) complex was further irradiated with red light to allow the transition from the closed to the open DTE configuration, producing the Tel22–DTE (1o’’) system. The latter CD spectrum presents a comparable shape to the one of Tel22–DTE (1o), although with an overall higher intensity. This indicates partial but not complete reversibility of the process. The CD spectra are shown in Figure 4b (red vs. orange curves).

To better quantify all the conformational changes upon complexation and irradiation, we used the algorithm reported in Ref. [22] to derive the secondary structure content of quadruplexes from their experimental CD spectra. In particular, a reference spectral library of several G4s of known structure was considered to fit CD profiles and quantitatively estimate the amounts of base steps in *anti*–*anti*, *syn*–*anti,* or *anti*–*syn* conformations, in *diagonal* or *lateral loops*, or in *other* sub-unities. A selection of experimental and theoretical best-fit profiles is reported in Appendix A., showing the robustness of the curve-fitting procedure. The most relevant components of the analysis are shown in Figure 4c, while the overall series of parameters is reported in Appendix A. Interestingly, upon complexation with DTE (1o), the *anti-anti* contribution is found to strongly decrease, suggesting less population of the parallel topology. At the same time, the percentage of *diagonal and lateral loops* increases while the *anti-syn* species decreases, indicating a concurrent rearrangement of hybrid and antiparallel conformations. The corresponding changes at the tertiary structural level, calculated using the code in Ref. [22], are shown in Appendix A. The further irradiation of the Tel22–DTE (1o) complex, producing a sample labeled as Tel22–DTE (1c’), causes a modification of both *diagonal and lateral loops* and *anti-syn* components that can explain the enhancement of the CD signal at 290 nm observed in Figure 4b. The percentage of all the components of Tel22–DTE (1o) is not fully restored upon the additional illumination with red light (see Tel22–DTE (1o’’) complex profile), indicating incomplete reversibility of the process.

#### 2.1.2. Tel22–TMPyP4 Complex

The impact of irradiation on the TMPyP4 ligand was also examined, as it is known to exhibit different characteristics after light exposure [37,40] with respect to the DTE ligand. The irradiation at 430 nm was chosen because it closely matches the main absorption band of TMPyP4, which peaks at around 422 nm and is associated with the Soret band, corresponding to the S0 to S2 transition [41]. We tested different exposure times for the TMPyP4 molecule to ensure that it did not undergo any degradation. Our findings indicated that, under our irradiation conditions (see Appendix A), the TMPyP4 absorption remains stable for exposure times up to 480 s. This information was then needed to define irradiation timing on the Tel22–TMPyP4 complex. Titration experiments showed that, as for DTE, the 1:2 DNA:ligand stoichiometric ratio optimizes the binding of TMPyP4 to Tel22 [37]. Moreover, in this case, the binding constant was found to be in the micromolar range. After being exposed to illumination, the Soret band, which has already shifted by 14 nm due to complexation with Tel22, further changes its peak position by an additional 1 nm. This shift is accompanied by a hypochromic effect of 20%, as shown in Appendix A.

CD spectra of Tel22 and Tel22–TMPyP4 are reported in Figure 5a. The first piece of evidence is that the peak at ∼270 nm, typical of the parallel-like G4 population, increases as a result of the complexation [37]. Then, the complex was irradiated with blue light. After exposure, porphyrin can generate singlet oxygens, thus oxidizing guanines at the exterior faces of the quadruplex scaffold. It was discovered that the major product of oxidation is 8-oxo-7,8-dihydroguanine (8-oxoG), which is susceptible to further oxidation, leading to the formation of spiroiminodihydantoin (Sp) and guanidinohydantoin [42]. As in the case of the Tel22–DTE (1c) complex, Tel22–TMPyP4 was illuminated at different irradiation times (i.t.), from 30 s to 480 s, and monitored by CD (Figure 5a). The progressive exposure produces an evident decrease in ellipticity, promoting conformational switching.

Starting from 120 s i.t., the CD signal strongly resembles those of 5′-TAGGGTTAXGGTTAGGGTTAGGGTT-3′ and 5′-TAGGGTTAGGXTTAGGGTTAGGGTT-3′ telomeric DNA sequences, where damage at the GGG triad has been introduced. In the structural formula of the modified DNA samples, lesion X stands for 8-oxoG or Sp products placed at the 5′ or 3′ ends, respectively [43]. These folds show maxima at 295 and 245 nm and a minimum at ∼265 nm, with a shape similar to the antiparallel-like topology. This result is highly likely because, upon exposure to radiation, porphyrin produces singlet oxygen species (^1^O_2_). In the interaction with G4s, these singlets preferentially oxidize guanines at the external quartets, sprouting numerous radicals and generating the major products 8-oxoG and Sp [44]. The comparison in Figure 5b between these oxidation products in G4s and Tel22–TMPyP4 irradiated for 480 s is particularly important since it enables localizing the damage in the Tel22 structure provoked by the irradiation-induced oxidation of the guanines. Our experiments also indicate that increasing the irradiation time up to 480 s causes the G4s to deteriorate until the tetrads progressively unstack. Indeed, the intensity of the Tel22–TMPyP4 i.t. 480 s spectrum is more than halved, and the shape is largely lost.

### 2.2. Conformational Changes upon Thermal Unfolding

In order to comprehend how the switching between secondary structure conformations might enable the regulation of telomere replication and to design small molecules that specifically bind to various conformers, it is crucial to understand the thermodynamic factors that affect the stability of different topologies. To investigate the thermodynamic impact of ligands on the stability of Tel22, CD experiments were carried out by varying the temperature until unfolding was reached. Measurements during the melting process give information on the stabilization/destabilization induced by the ligands and/or the irradiation.

#### 2.2.1. Thermal Unfolding of Tel22–DTE Complexes

The melting pathways of Tel22–DTE (1o) and Tel22–DTE (1c) were monitored through CD experiments, as shown in Figure 6a,b, respectively. The whole process was analyzed by means of the SVD method (details are given in Material and Methods, Section 4.2). In both cases, two intermediate states between folded and unfolded were detected, and the four-step process was described by a sequential unfolding model (F ⇌ I_1_ ⇌ I_2_ ⇌ U). A global nonlinear least-squares fitting on the amplitude vectors obtained from the SVD returned the values needed to reconstruct the spectra of the significant species (Figure 6c,d) and derive the thermodynamic parameters presented in Appendix A. For both cases, the melting temperature (T_m2_) is, within error, 7–8 degrees higher than that of Tel22 alone. The unfolded states have a typical CD feature of disordered polynucleotides with a very weak signal. The enthalpies of both steps are smaller in the case of Tel22–DTE (1o) compared with Tel22–DTE (1c), probably due to the fact that the G-tetrads in the complex with DTE in the open configuration are partially distorted, as suggested in Ref. [2]. Control experiments revealed significant thermal cyclo-reversion, which probably explains why the Tel22 complexes with the two isomers experience a similar melting pathway. A cartoon representation of the unfolding pathway is shown in Appendix A.

#### 2.2.2. Thermal Unfolding of Tel22–TMPyP4 Systems

It was observed that guanine oxidation alters the folding of telomeric quadruplexes in KCl solution [43]. The Tel22 melting pathway and thermostability are thus impacted by the interaction with TMPyP4 and subsequent irradiation of the complex. To quantify the oxidative effect on Tel22 thermal stability, the features of the Tel22–TMPyP4 complex ellipticity were monitored as a function of temperature both for the non-irradiated sample and for those exposed to blue light for 30 s, 120 s, and 480 s, respectively. All the corresponding CD profiles are displayed in Figure 7a. SVD analysis provides evidence of a three-step process (F ⇆ I ⇄ U), described by Equation (3) (see Section 4). The obtained thermodynamic parameters are reported in Appendix A, while the reconstructed significant species are in Figure 7b–e. We notice that, in the presence of TMPyP4, the melting temperature of Tel22 (T_m_ = 342 K [23]) is increased by about 3 degrees, as well as in the case of the Tel22–TMPyP4 irradiated for both 30 s and 120 s. This differs from what occurs in the Tel22–TMPyP4 sample irradiated for 480 s, where the melting temperature decreases by 2 degrees with respect to the unbound Tel22. As mentioned above, in this case, the DNA begins to lose its structure, and the complex becomes thermodynamically more unstable. Indeed, in addition to the decrease in the melting temperature, a reduction in the enthalpy between each step is observed.

By monitoring the whole melting path, it was observed that the intermediate states for Tel22–TMPyP4, Tel22–TMPyP4 i.t. 30 s and 120 s reported in Figure 8, although of gradually decreasing intensity, are very similar in shape (see Appendix A), and resemble a triple helical structure [45,46]. Interestingly, the triplex structure was suggested to be a possible pre-unfolding step for telomeric sequences [47,48,49,50]. Moreover, MD simulations and biophysical techniques revealed that triplexes are energetically feasible structures in the folding pathways of human telomeric type-1 and type-2 G4 conformations [31,46]. Furthermore, the complex irradiated for 480 s shows an intermediate state with a few differences from the others. In this case, the CD shape resembles that of a triplex with oxidized guanine in the central tetrad [43], as reported in Figure 8. This suggests that, while complexes irradiated for less time only oxidize the external tetrads, those irradiated for 480 s begin to oxidize even the guanines in the central tetrad. A schematic representation of the unfolding pathway is shown in Appendix A.

## 3. Discussions

In this work, we showed that the different responses of DTE and TMPyP4 photosensitive ligands to the stimulus of light affect in a distinct manner the structure and stability of Tel22.

We provided evidence that both isomers of DTE interact with Tel22, causing different quadruplex topological rearrangements. These structural changes are not completely restored upon irradiation. Even if the DTE isomerization is completely reversible both in the free and bound states with Tel22, as confirmed by UV-Vis absorption experiments, the switching process of the Tel22–DTE complex is not fully reversible, as evidenced by their corresponding CD spectra. Indeed, the blue irradiation of Tel22–DTE (1o) (giving rise to Tel22–DTE (1c’)) does not result in a spectrum identical to that obtained by complexing the Tel22 directly with DTE (1c), and vice versa. This suggests a possible impact of the complexation with DTE on the energy barriers between different Tel22 secondary structures. Indeed, in the K^+^ environment, Tel22 consists of a mixture of distinct conformers, which may interact in a diverse manner with DTE, thus generating slightly different secondary structure contents. Therefore, we hypothesize that the DTE molecule after irradiation of the complex (i.e., Tel22–DTE (1c’)) visits an energy landscape of quadruplex conformations different from the one available upon direct complexation (i.e., Tel22–DTE (1o) and Tel22–DTE (1c)). From a quantitative point of view, the analysis of CD spectra indicates that the combined effect of complexation and irradiation of Tel22–DTE systems, identified with an intensity increase of the band at 290 nm compared to that of Tel22 native structure (Figure 4a), results in a decrease of the *anti-anti* component and an increase of the *lateral* and *diagonal loops*. This suggests that the DTE ligand preferentially binds to the hybrid-like populations of the quadruplex mixture. Such a conjecture appears to be supported by the trend exhibited by the components of the tertiary structure (see Appendix A), i.e., the hybrid-like component shows a persistent growth in the complexed samples.

To be more general, a comparison with the literature can help. It was previously shown that in Na^+^ phosphate buffer, a different scenario occurs [2]. In this environment, the Tel22–DTE complex was found to undergo a completely reversible process; this result is confirmed here and shown in Appendix A (red vs. orange curves). Remarkably, Tel22 in Na^+^ assumes an antiparallel quadruplex conformation, contrary to the mixture of antiparallel and different hybrid topologies found in the K^+^ environment. DTE was also tested with the TAG_3_(TTAG_3_)_3_ telomeric sequence (Tel23) [2], which is recognized to have a hybrid-1 conformation in the K^+^ buffer [51], revealing a fully reversible process. These findings prove that the native conformation of telomeres plays a crucial role in determining their interaction properties. In light of the information gathered, it would be reasonable to hypothesize that the G4–DTE switching is reversible only when the native telomere assumes a unique conformation (e.g., hybrid-1 (pdb: 2JSM [51]) or antiparallel (pdb: 143D [27])), while reversibility is partially or completely lost when the telomere secondary structure corresponds to a mixture of conformations.

While our study is still at a speculative stage, it lays the groundwork for further investigation into the interaction between G4 and ligands. In particular, the present data can be corroborated through simulations, which could provide new insights into the different interactions of the ligand with a not-unique conformation G4 from the perspective of synthesizing properly designed ligands that can selectively interact with a specific quadruplex topology.

Moreover, irradiation and heating effects were compared, finding that the CD spectrum of the Tel22–DTE (1c’) sample when exposed to red light to switch the DTE isomer (giving rise to the Tel22–DTE (1o’’) profile shown in Figure 9a) is superimposable to the spectrum of its first intermediate state upon melting (T = 313 K, Appendix A). This evidence suggests that irradiation acts in a similar way to temperature in altering the equilibrium between the different Tel22 conformations. Figure 9a also shows that complexation, combined with both temperature and irradiation, promotes well-defined topological modifications. To give a quantitative estimate of the induced conformational changes on the Tel22 secondary structure, CD profiles were analyzed by means of the same algorithm used in Section 2.1.1. Accordingly, the most significant results are reported in Figure 9b, revealing the propensity of the *anti-anti* contribution to decrease, accompanied by a necessary rearrangement of the *lateral and diagonal loops* and *anti-syn* glycosidic bonds to reach a new equilibrium state.

Conversely to DTE, which, when exposed to radiation, is subject to photo-isomerization, TMPyP4 undergoes a chemical reaction, generating a different effect on the structure and stability of Tel22. Indeed, upon irradiation, porphyrins contribute to inducing the oxidation of guanines at the exterior faces when they form a complex with G4s [42]. The degree of such structural perturbation was controlled by varying the exposure time. We observed that if Tel22–TMPyP4 is irradiated for 480 s with blue light, it is subjected to a structural rearrangement that, at least partially, destroys the tetrads. These structural changes are similar to those obtained by introducing oxidized guanines in the G4 sequence [43]. In this case, gradual exposure to light is able to introduce irreversible modifications in Tel22, which can be useful for photodynamic applications. To confirm that the observed modifications were only due to the Tel22–TMPyP4 interaction and not to other light-related effects, the Tel22 alone and the Tel22–DTE (1c) complex were irradiated with blue light. Spectra at different irradiation times are presented in Appendix A, showing that neither Tel22 alone nor the Tel22–DTE (1c) complex displays significant modifications upon illumination.

Furthermore, after the complexation with these two ligands, we used temperature as an effective control parameter when visiting specific G4 intermediate structures. In a previous paper [23], we found out that the Tel22 explores two intermediate states before unfolding. Upon DTE complexation, the four-step thermal pathway is maintained, with DTE inducing a thermal stabilization of Tel22 of 7–8 degrees. We also discover that the F ⇌ I_1_ transition occurs in the two Tel22–DTE complexes at different temperatures, i.e., T_m1_ = 313 K for Tel22–DTE (1o) and T_m1_ = 317 K for Tel22–DTE (1c). On the other hand, the most interesting aspect of the SVD thermal analysis concerns the conformational diversity promoted by temperature in both samples. Indeed, the intermediate state I_1_ (Figure 6c,d, purple lines) tends to have a hybrid-like topology, while I_2_ (Figure 6c,d, pink lines) is much more parallel-like with respect to the corresponding folded structure (black line). The extent of the induced thermal changes was quantified and reported in Appendix A. Instead, three steps are involved in the thermal melting of Tel22–TMPyP4. SVD analysis gives the result that TMPyP4 is capable of stabilizing the Tel22 folded state by around 3–4 degrees (Appendix A). This behavior is maintained upon irradiation with blue light until 120 s, while the Tel22 thermodynamics are strongly affected for longer exposure times. In the case of i.t. 480 s, guanine oxidation was found to significantly alter the unfolding pathway of telomeric quadruplexes in a K^+^ environment and reduce their thermostability. Comparison with the literature allowed us to discover that, when the sample is irradiated for a long enough time (in our experiment, this plateau time corresponds to 480 s), oxidation does not only involve the extremal guanines but also the central tetrads of the quadruplex (Figure 8).

## 4. Materials and Methods

The oligonucleotide sequence AG_3_(TTAG_3_)_3_ was purchased from Eurogentec (Seraing, Belgium) and used without further purification. The lyophilized powder was dissolved in a 50 mM phosphate buffer, 150 mM KCl (substituted with NaCl for samples in a sodium environment), 0.3 mM EDTA, and pH = 7. The sample was heated to 95 °C for 5 min and then slowly cooled down to room temperature for ∼4 h. After this procedure, the samples were left at room temperature overnight. DNA concentration was determined from UV absorption measurements at 260 nm using a molar extinction coefficient of 228,500 M^−1^ cm^−1^ (data provided by Eurogentec). The DTE molecule was synthesized as described in Ref. [2] and dissolved in DMSO. 5,10,15,20-Tetrakis(1-methyl-4-pyridinio)porphyrin tetra(p-toluenesulfonate) (TMPyP4) was purchased from Merck KGaA (Darmstadt, Germany, cod. 36951-72-1) and dissolved in phosphate buffer.

All the measurements were performed considering Tel22 at 30 μM and complexed with DTE and TMPyP4 at a 1:2 [G4:ligand] stoichiometric ratio.

### 4.1. UV-Vis Absorption Spectroscopy

UV-vis absorption measures were performed using a Jasco V-750 spectrophotometer (Department of Physics and Geology, University of Perugia, Perugia, Italy) using a 1-mm path-length quartz cuvette. Spectra were recorded in the range of 200 to 800 nm.

### 4.2. CD Experiments

Circular dichroism experiments were carried out using a Jasco J810 spectropolarimeter (Department of Physics and Geology, University of Perugia, Perugia, Italy) on the same samples measured through UV-Vis spectroscopy using a 1-mm path-length quartz cuvette. Spectra were recorded in the range from 220 to 350 nm, with a scan speed of 50 nm/min. Melting measurements were performed by changing the temperature from 24 to 92 °C, with steps of 2 °C, for Tel22–DTE complexes and from 24 to 84 °C for Tel22–TMPyP4 systems.

### 4.3. SVD Details

The SVD is a method to factorize a matrix, D, into the product of three matrices, U, S, and V, i.e., D = U S V^T^, where V^T^ is the transpose of V.

The D matrix has as columns the CD experimental spectra at each temperature; the U matrix consists of the basis spectra, which combined are able to form the whole experimental dataset; and S is a diagonal matrix, where the numbers on the diagonal, the singular value, represent the weights of each component. The V matrix is made up of the amplitude vectors as a function of the temperature. The way of identifying the minimum number of spectral components able to reproduce the dataset is described in Refs. [45,52]. Basically, the magnitude and the relative variance of the singular values and the autocorrelation coefficients of the vectors of the U and the V matrices must be screened according to a certain acceptance/rejection criterion. As in the case of Tel22 alone, for Tel22–DTE complexes, a four-state model was found for the melting process: folded—intermediate 1—intermediate 2—unfolded (F ⇆ I_1_ ⇄ I_2_ ⇄ U). On the other hand, in the case of Tel22–TMPyP4 complexes, a state is suppressed, and a folded-intermediate-unfolded (F ⇆ I ⇄ U) melting pathway was identified [45]. Then, the significant V vectors were globally fitted to analytical expressions suitable for studying the thermodynamics of thermal unfolding. By analogy with proteins, changes in state functions were described in terms of the van’t Hoff equations [53], briefly mentioned below.

Let us recall that for a reversible process where a biomolecule passes from a native (F) to an unfolded (U) state (e.g., F ⇄ U) under the action of temperature, the following Equation holds to a good approximation:(1)$$\left[\varphi \right]=\frac{{\left[\varphi \right]}_{F}+K{\left[\varphi \right]}_{U}}{1+K}$$
where ${\left[\varphi \right]}_{F}$ and ${\left[\varphi \right]}_{U}$ are the variations of the physical observable for native (F) and unfolded (U) states, respectively, and [φ] that are detected in the transition region. The unfolding equilibrium constant K changes with temperature according to the van’t Hoff equation:(2)$$\left[K\left(T\right)\right]=exp\left[-\frac{∆H}{R}\left(\frac{1}{T}-\frac{1}{T_{m}}\right)\right]$$
where ∆H is the van’t Hoff unfolding enthalpy and T_m_ the denaturation temperature. As G4s are generally characterized by multistep thermal paths, Equation (1) needs to be adapted on a case-by-case basis. In Ref. [45], several mechanisms were proposed and have to be tested on experimental datasets.

Equations (3) and (4) were used for Tel22–DTE and Tel22–TMPyP4, respectively.
(3)$$s(T)=\frac{S_{U}e^{-\frac{{dH}_{1}}{R}\left(\frac{1}{T_{m1}}-\frac{1}{T}\right)-\frac{{dH}_{2}}{R}\left(\frac{1}{T_{m2}}-\frac{1}{T}\right)}+S_{I}e^{-\frac{dH_{1}}{R}\left(\frac{1}{T_{m2}}-\frac{1}{T}\right)}{+S}_{F}}{e^{-\frac{{dH}_{1}}{R}\left(\frac{1}{T_{m1}}-\frac{1}{T}\right)-\frac{dH_{2}}{R}\left(\frac{1}{T_{m2}}-\frac{1}{T}\right)}+e^{-\frac{dH_{1}}{R}\left(\frac{1}{T_{m2}}-\frac{1}{T}\right)}+1}$$
(4)$$s(T)=\frac{{S_{U}e^{-\frac{dH1}{R}\left(\frac{1}{T_{m1}}-\frac{1}{T}\right)-\frac{dH2}{R}\left(\frac{1}{T_{m2}}-\frac{1}{T}\right)-\frac{dH3}{R}\left(\frac{1}{T_{m3}}-\frac{1}{T}\right)}+S_{2}}_{}e^{-\frac{dH1}{R}\left(\frac{1}{T_{m1}}-\frac{1}{T}\right)-\frac{dH2}{R}\left(\frac{1}{T_{m2}}-\frac{1}{T}\right)}+S_{I}e^{-\frac{dH_{1}}{R}\left(\frac{1}{T_{m2}}-\frac{1}{T}\right)}{+S}_{F}}{{e^{-\frac{dH1}{R}\left(\frac{1}{T_{m1}}-\frac{1}{T}\right)-\frac{dH2}{R}\left(\frac{1}{T_{m2}}-\frac{1}{T}\right)-\frac{dH3}{R}\left(\frac{1}{T_{m3}}-\frac{1}{T}\right)}+e^{-\frac{dH1}{R}\left(\frac{1}{T_{m1}}-\frac{1}{T}\right)-\frac{dH2}{R}\left(\frac{1}{T_{m2}}-\frac{1}{T}\right)}}^{}+e^{-\frac{dH_{1}}{R}\left(\frac{1}{T_{m2}}-\frac{1}{T}\right)}+1}$$
where dH_i_ = dH_(folding)_ for step i; T_mi_ = mid-point temperature for step i (with i = 1 corresponding to the F ↦ I step and i = 2 to the I ↦ U step); S_F_ = optical signal for folded conformers; S_I_ = optical signal for the intermediate species; S_U_ = optical signal of the unfolded ensemble; R = 1.987 cal K^−1^ mol^−1^.

## 5. Conclusions

In this work, we studied the conformational changes, before and after irradiation with selective wavelengths, of the Tel22 human telomeric sequence complexed with two photosensitive ligands, DTE and TMPyP4, which respond differently to visible light.

We discovered that, even though DTE switches in a completely reversible manner between its two isomers, the reversibility of the Tel22–DTE system is not observed. This can be explained by considering that Tel22 is a mixture of secondary structure topologies. While the interaction between DTE and hybrid 1/antiparallel telomeric G4 sequences is already known, that with other conformations is still unresolved, and thus it probably compromises the reversibility.

Future studies could be designed to investigate DTE reversibility with other G4 sequences that assume unique topologies, proving which are the fundamental interaction properties that guarantee/affect the reversibility. Those outputs may be particularly insightful for the design of selectively interacting small molecules in different telomeric conformations. The other aspect of the study focused on TMPyP4, which produces singlet oxygen (^1^O_2_) upon exposure to radiation and, when bound to Tel22, causes guanine oxidation. We demonstrated that the oxidation level can be tuned by the irradiation time, a crucial piece of proof supporting phototherapeutic methods.

The effect of the temperature on the unfolding was also investigated, underlying the occurrence of multi-step melting pathways and highlighting, for both complexes, the presence of intermediate states (whose topological details can be quantified) before the quadruplex unfolding.

Overall, by analyzing the cases of Tel22–DTE and Tel22–TMPyP4 complexes, we showed that changes in the secondary structure of human telomeric quadruplexes interacting with the two photosensitive ligands may be fine-tuned on demand with temperature and light. Compared to other conformational controllers (e.g., the addition of crowders/salts), using light is, however, a better tool for applicative purposes. In fact, the timing, location, and intensity of light can be controlled readily, opening the way for the application of G4-ligand complexes as responsive systems in many fields. Toward this end, a set of telomeric sequences can be tested against other photoswitchable ligands, such as azobenzenes, arylstilbazolium, or stiff-stilbene [11], in order to explore their interaction properties. For future applications, light also offers a number of benefits over chemical stimuli for the regulation of G4 formation/topology in vivo as a potential therapeutic strategy.

## Figures and Tables

**Figure 1 ijms-24-09090-f001:**
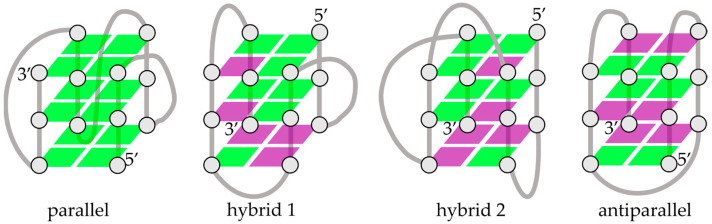
Schematic representation of the G4 folded structures in the main topologies. The green and purple squares represent the guanine bases in the corresponding *anti* and *syn* conformations, respectively.

**Figure 2 ijms-24-09090-f002:**
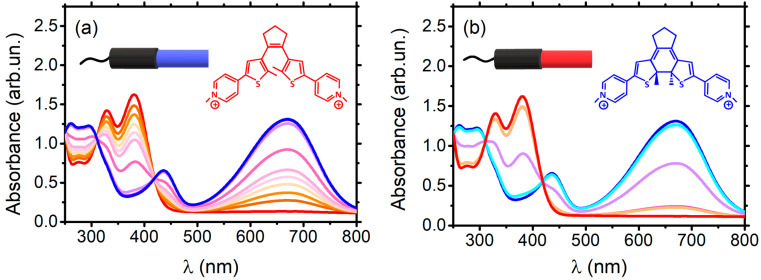
UV-Vis absorption spectra at increasing irradiation times of (**a**) DTE open to closed (1o)→(1c) (from red to blue) and (**b**) DTE closed to open (1c)→(1o) (from blue to red). Spectra related to the process (1o)→(1c) are obtained irradiating the sample from 0 to 10 min with 430 nm, while those related to (1c)→(1o) from 0 to 1.5 h with 660 nm light.

**Figure 3 ijms-24-09090-f003:**
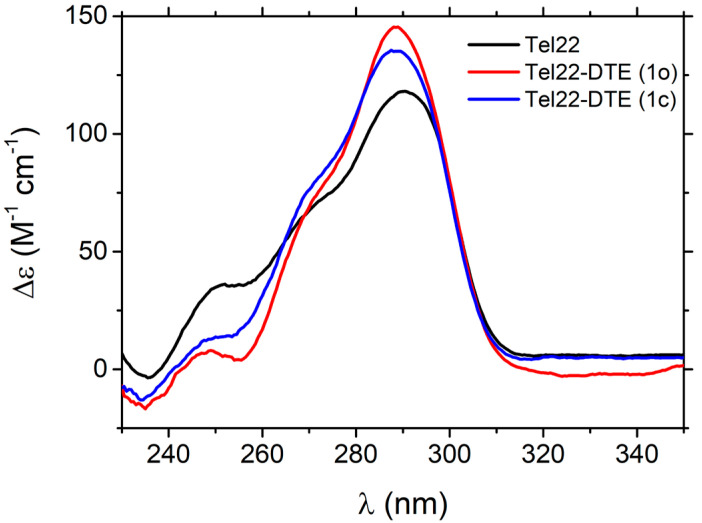
CD spectra of Tel22 (black), Tel22–DTE (1o) (red), and Tel22–DTE (1c) (blue). Tel22 was at 30 μM and DTE (both configurations) at 60 μM.

**Figure 4 ijms-24-09090-f004:**
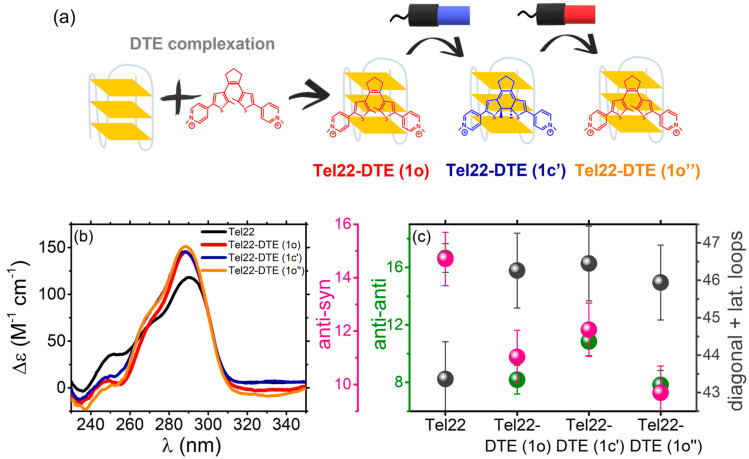
(**a**) Schematic representation of Tel22–DTE (1o) complexation and irradiation. (**b**) CD spectra of Tel22 (black), Tel22–DTE (1o) (red), Tel22–DTE (1c’) (dark blue), corresponding to a Tel22–DTE (1o) sample after illumination at 430 nm for 10 min, and Tel22–DTE (1o’’) (orange), obtaining after the irradiation of Tel22–DTE (1c’) at 660 nm for 1.5 h. (**c**) Percentage of the main secondary structure components obtained by deconvolving the CD spectra through the method proposed in Ref. [22]. whit *anti-syn* components in pink, *anti-anti* in green and *diagonal and lateral loops* in grey.Tel22 was at 30 μM and DTE (in both configurations) at 60 μM.

**Figure 5 ijms-24-09090-f005:**
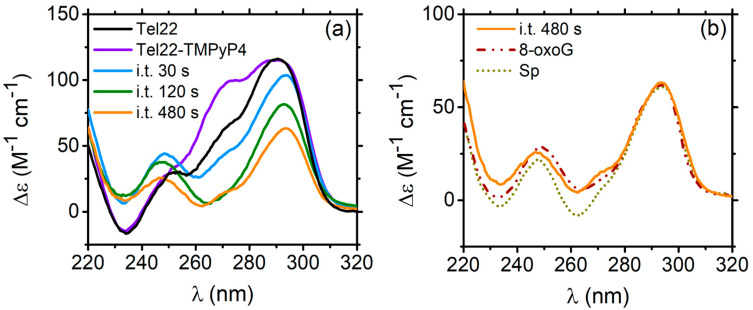
(**a**) CD spectra of Tel22 (black), Tel22–TMPyP4 (purple), Tel22–TMPyP4 i.t. 30 s (light blue), Tel22–TMPyP4 i.t. 120 s (green), and Tel22–TMPyP4 i.t. 480 s (orange). (**b**) CD spectra of Tel22–TMPyP4 i.t. 480 s (orange), 8-oxoG (brown, dot-dashed line), and Sp (dark yellow, dotted line). The dotted CD spectra are taken from Ref. [43] and properly rescaled for comparison.

**Figure 6 ijms-24-09090-f006:**
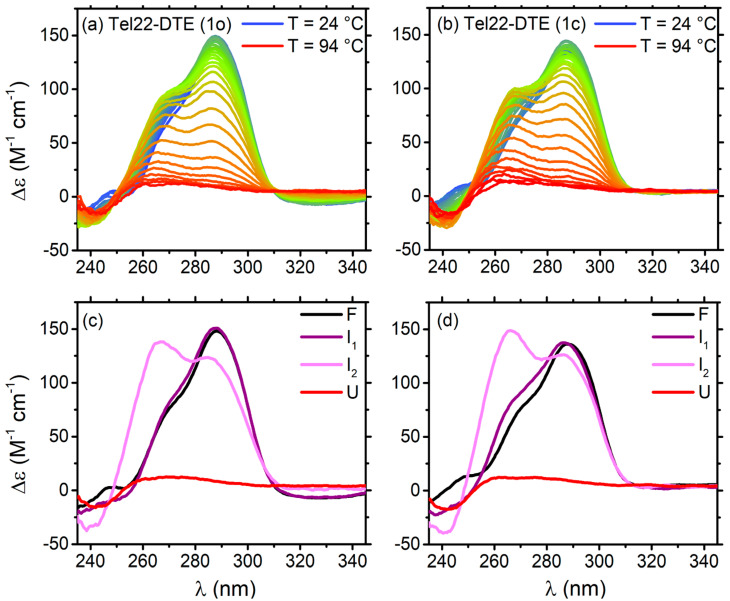
CD melting spectra of: (**a**) Tel22–DTE (1o) and (**b**) Tel22–DTE (1c). Reconstructed spectra of significant species (F, I_1_, I_2_, U): (**c**) for Tel22–DTE (1c) and (**d**) for Tel22–DTE (1o). Tel22 was at 30 μM and DTE (both configurations) at 60 μM.

**Figure 7 ijms-24-09090-f007:**
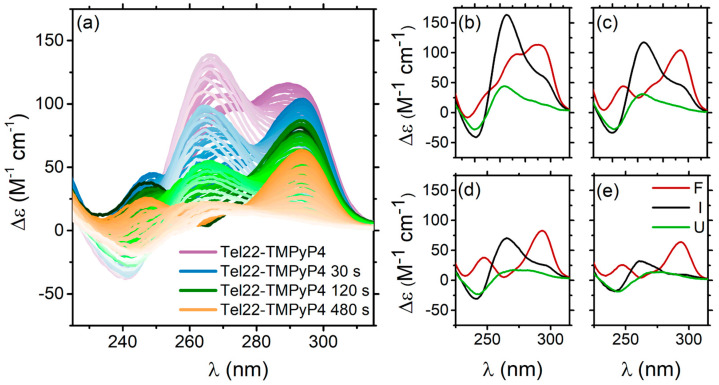
(**a**) CD spectra of the Tel22–TMPyP4 complex as a function of temperature for different irradiation times. Reconstructed spectra (F, I, and U) from SVD analysis of Tel22–TMPyP4 complex irradiated with 430 nm wavelength for: (**b**) Tel22–TMPyP4 0 s, (**c**) Tel22–TMPyP4 30 s, (**d**) Tel22–TMPyP4 120 s, and (**e**) Tel22–TMPyP4 480 s.

**Figure 8 ijms-24-09090-f008:**
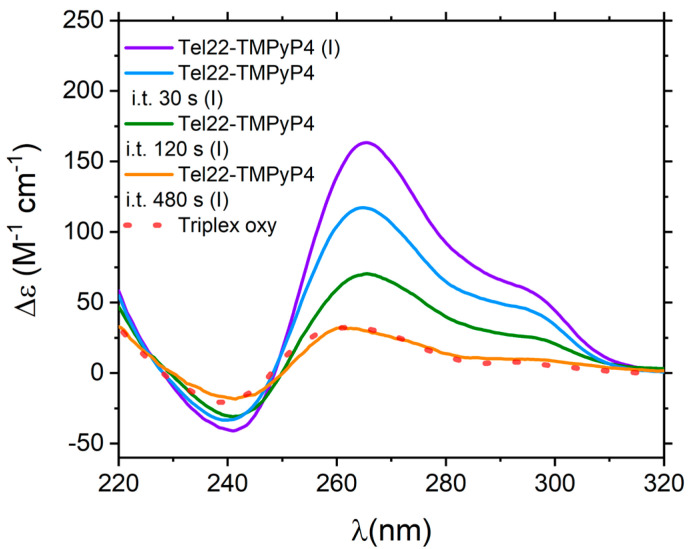
CD spectra of the intermediates reconstructed via SVD. Tel22–TMPyP4 (violet), Tel22–TMPyP4 i.t. 30 s (light blue), Tel22–TMPyP4 i.t. 120 s (green), Tel22–TMPyP4 i.t. 480 s (orange). Dashed line represents the CD spectrum of a triplex that is a result of telomeric G4 with oxidized guanine in the central tetrads (taken from Ref. [43]).

**Figure 9 ijms-24-09090-f009:**
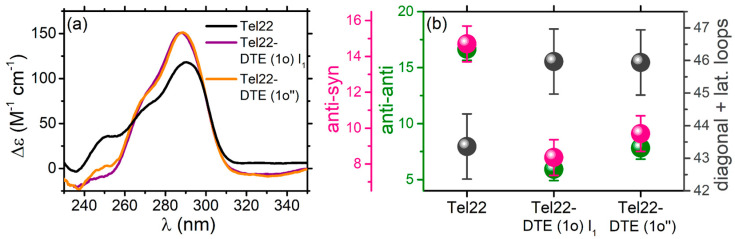
(**a**) CD spectra of Tel22, of intermediate 1 of the Tel22–DTE (1o) complex, and of Tel22–DTE (1o’’), which were obtained from the irradiation with 660 nm of the Tel22–DTE (1c’). (**b**) Percentage of the main secondary structure components obtained by deconvolving the CD spectra through the method proposed in Ref. [22]. Tel22 was at 30 μM and DTE (both configurations) at 60 μM.

## Data Availability

Not applicable.

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
