# Peer review of "Stability of Human Telomeric G-Quadruplexes Complexed with Photosensitive Ligands and Irradiated with Visible Light"

_ijms, 2023, doi:10.3390/ijms24109090_

Round 1

Reviewer 1 Report

See attached review.

Reviewer 2 Report

In this manuscript entitled “Stability of human telomeric G-quadruplexes complexed with photosensitive ligands and irradiated with visible light”, Libera et al studied effects of photosensitive ligands, DTE and TMPyP4” on structure of human telomeric DNA and its thermal stability. It has been reported that some photosensitizers are able to target G-quadruplexes of DNA and RNA, leading to inhibition of expression of a certain gene (Xodo, Kawauchi, and others). On the other hand, it is not easy to understand a purpose of this study for a broad reader of the journal. Please explain why the authors studied effects of the photosensitizers on the conformational change and thermal stability of a human telomeric DNA. In the same way, please explain why the authors used this sequence, which shows very polymorphic structure depending on an environmental condition. Moreover, this reviewer suggests some improvements as the follows:

1. Please show chemical structure of the ligands to help the readers for understanding.

2. Before the detail study of the complex of the ligands and G4, please evaluate some binding parameters such as a binding constant.

3. The changes in the CD spectra shown in Figure 2 (b) and Figure 3, are small. Even the peaks and shoulders have been identified as shown in the reference 22, it is still required to confirm the conformation of the G4s with the ligands before and after photoirradiation by use of independent method, such as NMR.

Round 2

Reviewer 1 Report

The authors essentially addressed all the major issues raised by myself and the other reviewer. I have no additional concern and in my opinion this article can be accepted for publication.

Reviewer 2 Report

This manuscript has been improved according to the reviewers suggestion. This reviewer is now glad to suggest publication of this manuscript in IJMS.